# Chemogenetic Inhibition of Prefrontal Cortex Ameliorates Autism-Like Social Deficits and Absence-Like Seizures in a Gene-Trap *Ash1l* Haploinsufficiency Mouse Model

**DOI:** 10.3390/genes15121619

**Published:** 2024-12-18

**Authors:** Kaijie Ma, Kylee McDaniel, Daoqi Zhang, Maria Webb, Luye Qin

**Affiliations:** 1Division of Basic Biomedical Sciences, Sanford School of Medicine, University of South Dakota, Vermillion, SD 57069, USA; kaijie.ma@usd.edu (K.M.);; 2Department of Biotechnology, Mount Marty University, Yankton, SD 57078, USA; kylmcd001@mountmarty.edu; 3School of Health Sciences, University of South Dakota, Vermillion, SD 57069, USA

**Keywords:** autism spectrum disorder, *ASH1L*, social behaviors, seizures, neural excitability, prefrontal cortex

## Abstract

Background: *ASH1L* (absent, small, or homeotic-like 1), a histone methyltransferase, has been identified as a high-risk gene for autism spectrum disorder (ASD). We previously showed that postnatal *Ash1l* severe deficiency in the prefrontal cortex (PFC) of male and female mice caused seizures. However, the synaptic mechanisms underlying autism-like social deficits and seizures need to be elucidated. Objective: The goal of this study is to characterize the behavioral deficits and reveal the synaptic mechanisms in an *Ash1l* haploinsufficiency mouse model using a targeted gene-trap knockout (gtKO) strategy. Method: A series of behavioral tests were used to examine behavioral deficits. Electrophysiological and chemogenetic approaches were used to examine and manipulate the excitability of pyramidal neurons in the PFC of Ash1l^+/GT^ mice. Results: Ash1l^+/GT^ mice displayed social deficits, increased self-grooming, and cognitive impairments. Epileptiform discharges were found on electroencephalograms (EEGs) of Ash1l^+/GT^ mice, indicating absence-like seizures. *Ash1l* haploinsufficiency increased the susceptibility for convulsive seizures when Ash1l^+/GT^ mice were challenged by pentylenetetrazole (PTZ, a competitive GABA_A_ receptor antagonist). Whole-cell patch-clamp recordings showed that *Ash1l* haploinsufficiency increased the excitability of pyramidal neurons in the PFC by altering intrinsic neuronal properties, enhancing glutamatergic synaptic transmission, and diminishing GABAergic synaptic inhibition. Chemogenetic inhibition of pyramidal neurons in the PFC of Ash1l^+/GT^ mice ameliorated autism-like social deficits and abolished absence-like seizures. Conclusions: We demonstrated that increased neural activity in the PFC contributed to the autism-like social deficits and absence-like seizures in Ash1l^+/GT^ mice, which provides novel insights into the therapeutic strategies for patients with *ASH1L*-associated ASD and epilepsy.

## 1. Introduction

Autism spectrum disorder (ASD) is a group of neurodevelopmental disorders with strong genetic heterogeneity. Epilepsy is a major comorbidity of children with ASD. Up to 80% of children with ASD show EEG abnormality (epileptiform discharges). Up to 30% of children with ASD have epilepsy showing seizures, and 30% of them are antiepileptic drug-resistant [1,2,3,4,5,6]. Functional characterization of the risk genetic factors causing epilepsy and ASD will lead to novel therapeutic strategies to control seizures and ameliorate autism-like behavioral deficits [7,8,9].

*ASH1L* (absent, small, or homeotic-like 1), a histone methyltransferase, is a high-risk gene for ASD based on identified high phenotypic penetration disease-causing loss-of-function (LOF) mutations (Simons Foundation Autism Research Initiative, SFARI) [7,8,9]. Individuals harboring *ASH1L* LOF variants display seizures, social deficits, and other neurodevelopmental deficits [7,9,10,11,12,13,14,15]. *Ash1l* null mice died perinatally, suggesting its critical role in brain development. *Ash1l* haploinsufficiency or conditional knockout *Ash1l* in neural progenitor cells induced autism-like behavioral deficits [16,17]. Our previous studies showed that postnatal *Ash1l* severe deficiency (80% reduction) in the prefrontal cortex (PFC) of juvenile or early adolescent male and female mice caused seizures [18]. Others and our studies in preclinic mouse models confirmed the causal role of *ASH1L* in ASD and epilepsy.

Disrupted synaptic excitatory and inhibitory (E/I) balance caused by dysfunctions of synaptic genes and/or ion channels have been recognized as a key site of pathogenesis in epilepsy and ASD [19,20,21,22,23,24,25,26,27]. PFC is a critical hub providing “top-down” control for social behaviors and cognition [28,29,30,31], one of the key brain regions impaired in children with ASD. The human developmental trajectory of PFC showed that the expressions of synaptic genes and density peak at 3.5–10 years of age [32,33,34], a critical time window of phenotypic manifestations, diagnosis, and treatment for children with ASD, intellectual disability, and epilepsy. *ASH1L* is highly expressed in excitatory and inhibitory neuronal lineages [8], which indicates germline *ASH1L* haploinsufficiency impairs postnatal synaptic functions. However, the underlying synaptic mechanisms of how *ASH1L* haploinsufficiency contributes to autism-related social deficits and seizures are largely unknown.

In this study, we characterized the impact of *Ash1l* haploinsufficiency on autism-related behaviors and seizures in a mouse model generated by a gene-trap (GT) knockout strategy [35]. *Ash1l* haploinsufficiency caused autism-like behavioral deficits, absence-like seizures, and increased the susceptibility for PTZ-induced convulsive seizures. *Ash1l* haploinsufficiency increased the excitability of pyramidal neurons by altering intrinsic neuronal properties and synaptic inputs.

Designer receptors exclusively activated by designer drugs (DREADDs) are engineered G-protein coupled receptors that can be activated by otherwise inert drug-like ligands, such as clozapine-N-oxide (CNO). Activation of DREADD encoded Gq-coupled M3-muscarinic receptor (hM3Dq) induces neuronal burst firing, while activation of Gi-coupled M4-muscarinic receptor (hM4Di) causes neuronal silencing [36,37]. Chemogenetic inhibition of pyramidal neurons in the PFC of Ash1l^+/GT^ mice ameliorated autism-like social deficits and abolished absence-like seizures. This study identified synaptic mechanisms underlie the autism-like social deficits and epilepsy in Ash1l^+/GT^ mice, which provides potential therapeutic development for patients carrying *ASH1L* mutations.

## 2. Materials and Methods

### 2.1. Animal Care and Husbandry

The use of animals and procedures performed were approved by the Institutional Animal Care and Use Committee of Sanford School of Medicine, University of South Dakota. Ash1l^+/GT^ mice (Jackson Laboratory stock #028220) were generated as previously described [35]. Animals were group-housed (n = 4–5 mice) in standard cages and were kept on a 12 h light–dark cycle in a temperature-controlled room. Food and water were available ad libitum. Male and female heterozygous Ash1l^+/GT^ mice and sex- and age- matched Ash1l^+/+^ mice were included in the experiments, which were derived from heterozygous Ash1l^+/GT^ breeding pairs. All behavioral assays were performed when mice were 6–8 weeks old. Experiments were carried out by investigators in a blinded fashion (with no knowledge of genotypes).

### 2.2. Behavioral Tests

A series of behavioral tests were performed as in our previous studies [29,38], including open field test, rotarod test, social preference test, self-grooming, Barnes maze test, and novel object recognition (NOR) test.

### 2.3. LacZ (β-galactosidase) Staining

Mice were anesthetized and perfused with PBS, followed by 4% paraformaldehyde (PFA). After being post-fixed in 4% PFA overnight, the brains were cut into 100 μm sagittal slices. The LacZ (β-galactosidase) staining was carried out using the β-galactosidase staining kit (Cell Signaling Technology, #9860, Danvers, MA, USA) according to the manufacturer’s instructions with minor modifications. In brief, floating slices were incubated with freshly prepared staining solution at 37 °C in a humid chamber until color developed. Stained slices were washed with PBS, dehydrated with a series of ethanol, cleared with xylene, and covered with Permount medium. Images were taken under a stereotactic microscope.

### 2.4. Pentylenetetrazol (PTZ, a Competitive GABA_A_ Receptor Antagonist) Administration

To test the susceptibility for convulsive seizures, mice were injected with a low-dose PTZ (Sigma, St. Louis, MO, USA, #P6500, 35 mg/kg, single dose, daily, intraperitoneal injection) for 3 days. Immediately following PTZ injection, mice were monitored for seizure activity for 30 min by a lab member with no prior knowledge of genotypes. The quantification of seizure severity was based on published scoring criteria with a modified version of the Racine scale [39,40,41].

### 2.5. EEG

EEG recordings were performed as described in our previous studies [18,42]. EEG signals were continuously recorded at a sampling rate of 20 KHz (Intan 512ch Recording Controller, Part #C3004, Intan Technologies, Los Angeles, CA, USA), and separated by a band-pass filter at 0.1–100 Hz by Offline sorting software V4 and analyzed by Neuroexplorer V5.0 (Plexon, Dallas, TX, USA).

### 2.6. Brain Slice Preparation

Coronal brain slices containing PFC were prepared from male and female Ash1l^+/+^ and Ash1l^+/GT^ mice (6–8 weeks old) as described previously [18,28,29].

### 2.7. Whole-Cell Patch-Clamp Recordings

Whole-cell patch-clamp recordings were performed with a multi-Clamp 700B amplifier (Molecular Devices, San Jose, CA, USA), and data were acquired using pClamp 11.2 software, filtered at 1 kHz and sampling rate at 10 kHz with an Axon Digidata 1550B plus HumSilencer digitizer (Molecular Devices) as previously described [18,28,29,43,44].

### 2.8. Quantitative Real-Time PCR

Quantitative real-time PCR was performed as described in our previous studies [18,28,29]. In brief, GAPDH was used as the housekeeping gene for quantitation of the expression of target genes in samples from Ash1l^+/+^ and Ash1l^+/GT^ mice. Fold change = 2^−Δ(ΔCT)^, where ΔCT = CT (target genes) − CT(GAPDH), and Δ(ΔCT) = ΔCT (Ash1l^+/GT^ mice) − ΔCT (Ash1l^+/+^ mice). CT (threshold cycle) was defined as the fractional cycle number at which the fluorescence reaches 10x of the standard deviation of the baseline. Primers for all target genes were listed in Table 1.

### 2.9. Viral Vectors and Animal Surgery

The AAV9-CaMKIIα-hM4D(Gi)-mCherry were obtained from Addgene (Watertown, MA, USA). The virus (1 μL) was bilaterally injected into the medial PFC (1.9 mm anterior to the bregma, 0.25 mm lateral, and 2.0 mm deep) to infect pyramidal neurons, as we described before [18,28]. Animals were used for experiments 2–3 weeks later. CNO (Tocris, Minneapolis, MN, USA, #6329) or saline injection (i.p.) was given 1 h before the start of behavioral testing or EEG recordings. Behavioral testing or EEG recordings were performed at 1–6 h after injection of CNO or saline in Gi-GREADD-infected mice.

### 2.10. Western Blotting

Nuclear extracts from mouse PFC punches were prepared and Western blotting experiments were performed as described in our previous studies [18,29]. Western blotting was performed with antibodies against Ash1l (1:1000, LSBio, Newark, CA, USA, #LS-B11718) and Histone 3 (1:500, Cell Signaling, Danvers, MA, USA, #4499).

### 2.11. Statistical Analysis

To detect behavioral differences in mice, sample size was calculated based on predicting detectable differences to reach a power of 0.80 at a significance level of 0.05 by running power analyses in G*Power software 3.1.9.7. Statistical comparisons were performed using GraphPad software Prism 7.0 (GraphPad Software, La Jolla, CA, USA). Differences between two groups were assessed with unpaired two-tailed Student’s *t*-test. Differences between more than two groups were assessed with one-way or two-way ANOVA, followed by post hoc Bonferroni tests for multiple comparisons. Data were presented as mean ± SEM.

## 3. Results

### 3.1. Validation of Gene-Trap Knockout of Ash1l in the Brain

The *Ash1l* haploinsufficiency mouse model was generated with a targeted gene-trap knockout strategy [35]. As shown in Figure 1A, a gene-trap cassette containing the β-geo reporter gene (β-galactosidase and neomycin resistance fusion gene) was inserted into the intron 1 of the *Ash1l* gene, thus “trapping” the splicing of exon 1 of *Ash1l* to produce a LacZ fusion transcript and truncating the wild-type transcript. Representative genotyping was shown in Figure 1B. Consistent with the previous report, heterozygous Ash1l^+/GT^ mice had normal growth. However, homozygous Ash1l^GT/GT^ mice had smaller body weights and died within 1–3 weeks after birth [35]. Thus, male and female Ash1l^+/+^ and Ash1l^+/GT^ mice (6–8 weeks old) were used in this study. To identify the transcripts containing Exon 1 spliced to the trapping cassette containing LacZ, we performed β-gal staining to show the distribution of TRAP in Ash1l^+/GT^ mice, which confirmed that gene trapping was functioning in the brain (Figure 1C). The knockout efficacy of *Ash1l* in the brain was determined by measuring the mRNA and protein levels of *Ash1l* in the prefrontal cortex. Compared with Ash1l^+/+^ mice, the mRNA and protein levels of *Ash1l* were significantly decreased in male and female Ash1l^+/GT^ mice (Figure 1D,E) (mRNA: Ash1l^+/+^: 1 ± 0.04; Ash1l^+/GT^: 0.59 ± 0.03, n = 12 mice (6 males and 6 females)/group, *** *p* < 0.001, t_22_ = 7.8, unpaired two-tailed *t*-test. Protein: Ash1l^+/+^:1 ± 0.09; Ash1l^+/GT^: 0.61 ± 0.04, n = 8 mice (4 males and 4 females)/group, ** *p* < 0.01, t_14_ = 3.9, unpaired two-tailed *t*-test).

### 3.2. Ash1l Haploinsufficiency Causes Autism-Like Behavioral Deficits in Male and Female Mice

To determine the impact of *Ash1l* haploinsufficiency on social behaviors, Ash1l^+/GT^ mice and sex- and age- matched Ash1l^+/+^ mice were subjected to the three-chamber social interaction assay [28,29]. As shown in Figure 2A–C, Ash1l^+/+^ mice spent significantly more time exploring the social stimulus over the non-social object, while Ash1l^+/GT^ mice displayed reduced preference for the social stimulus (Ash1l^+/+^: social: 156.2 ± 8.4 s, nonsocial: 58.6 ± 4.9 s, n = 15 mice (7 males and 8 females); Ash1l^+/GT^: social: 94.5 ± 6.8 s, nonsocial: 87.6 ± 5.4 s, n = 14 mice (7 males and 7 females); *F*
_Genotype (1, 54)_ = 6.3, *p* = 0.015; *F*
_Soc vs. NS (1, 54)_ = 63.8, *p* < 0.001, *F*
_Interaction (1, 54)_ = 48, *p* < 0.001; two-way ANOVA). Consistently, Ash1l^+/GT^ displayed significantly reduced social preference index, compared with Ash1l^+/+^ mice (Ash1l^+/+^ mice: 45.3% ± 3.8%, n = 15 mice (7 males and 8 females); Ash1l^+/GT^ mice: 8.7% ± 4.3%, n = 14 mice (7 males and 7 females); *p* < 0.001, t_27_ = 6.4, unpaired two-tailed *t*-test). Ash1l^+/GT^ mice spent significantly more time in self-grooming (Ash1l^+/+^ mice: 15.2 ± 2.6 s, n = 15 mice (7 males and 8 females); Ash1l^+/GT^ mice: 25.1 ± 2.3 s, n = 14 mice (7 males and 7 females); ** *p* < 0.01, t_27_ = 2.8, unpaired two-tailed *t*-test, Figure 2D), which has been used as an indication of compulsive and repetitive behavior in human ASD patients [45]. These data indicated that male and female Ash1l^+/GT^ mice exhibited social deficits and repetitive behaviors, the two core behavioral features of ASD.

To assess the impact of *Ash1l* haploinsufficiency on cognition, NOR tests were used to examine whether *Ash1l* haploinsufficiency could impair object recognition memory. As shown in Figure 2E, Ash1l^+/GT^ mice had similar discrimination index with Ash1l^+/+^, indicating *Ash1l* haploinsufficiency had no effect on the object recognition memory. To determine whether *Ash1l* haploinsufficiency affects spatial memory, we performed a Barnes maze test [46,47]. Compared with Ash1l^+/+^ mice, Ash1l^+/GT^ mice displayed significantly lower spatial memory index (T1/T2) (Ash1l^+/+^: 0.63 ± 0.04, n = 15 mice (7 males and 8 females); Ash1l^+/GT^: 0.4 ± 0.03, n = 14 mice (7 males and 7 females); *** *p* < 0.001, t_27_ = 4.5, unpaired two-tailed *t*-test). These results suggested that *Ash1l* haploinsufficiency impaired spatial memory. Ash1l^+/+^ and Ash1l^+/GT^ mice displayed similar time spent at the center in the open field test. The similar distance traveled in the open field test and latency to fall during the rotarod test demonstrated that *Ash1l* haploinsufficiency had no effect on anxiety-like behaviors, locomotion activity, and movement coordination (Figure 2I–L).

### 3.3. Ash1l Haploinsufficiency Causes Absence-Like Seizures and Increases the Susceptibility for Convulsive Seizures

Among ASD children with epilepsy, about 80% of them appear to have seizures in early childhood, adolescence, and adulthood [5,6]. However, no spontaneous convulsive seizures were observed in Ash1l^+/GT^ mice at 6–8 weeks old, a time window for humans presenting seizures. Given that up to 80% of children with ASD show epileptiform discharges on EEG, we conducted EEG recordings to measure the neural activity in the PFC of freely behaving Ash1l^+/+^ and Ash1l^+/GT^ mice. Ash1l^+/GT^ mice showed epileptiform discharges (an interictal marker of absence seizures) on EEG and accompanied with a brief cessation of movement, which demonstrated that *Ash1l* haploinsufficiency induced absence-like seizures (Figure 3A). The percentage total power of delta frequency was significantly increased, and the gamma frequency was significantly decreased in Ash1l^+/GT^ mice (Figure 3B). To further determine whether *Ash1l* haploinsufficiency increases the susceptibility for convulsive seizures, we challenged Ash1l^+/+^ and Ash1l^+/GT^ mice (6–8 weeks old) by systemic administration of low-dose PTZ (35 mg/kg, single dose, daily, intraperitoneal injection.) for 3 days. The seizure scores were measured 30 min after PTZ administration based on the Racine seizure-behavior scoring paradigm [39,40,41]. A low-dose PTZ injection induced immobilization, head nodding, and partial myoclonus in Ash1l^+/GT^ mice and minor effects on Ash1l^+/+^ mice with random immobilization. The seizure scores were significantly higher in Ash1l^+/GT^ mice, compared with Ash1l^+/+^ mice (*F*
_Genotype (1, 38)_ = 38.1, *p* < 0.001, *F*
_Day (2, 76)_ = 31.6, *p* < 0.001, *F*
_Interaction (2, 76)_ = 8.5, *p* < 0.001, two-way ANOVA, n = 20 mice (10 males and 10 females)/group.) These results suggested that *Ash1l* haploinsufficiency caused absence-like seizures and increased the susceptibility for PTZ-induced convulsive seizures in both genders.

### 3.4. Ash1l Haploinsufficiency Increases the Excitability of Pyramidal Neurons in the PFC

To determine the synaptic mechanisms underlying the autism-like social deficits and absence-like seizures in Ash1l^+/GT^ mice, we performed whole-cell patch-clamp recordings in PFC slices. We first examined the impact of *Ash1l* haploinsufficiency on the intrinsic excitability of pyramidal neurons in the layer V of PFC, which showed the clearest deficits in autistic children [48]. As shown in Figure 4A,B, the spike number of eAPs was significantly higher in Ash1l^+/GT^ mice, compared with Ash1l^+/+^ mice (*F*
_Genotype (1, 38)_ = 56.9, *p* < 0.0001, *F*
_Intensity (14, 532)_ = 250.5, *p* < 0.001, *F*
_Genotype and Intensity interaction (14, 532)_ = 22.2, *p* < 0.001, two-way ANOVA). Next, we examined the action potential properties of pyramidal neurons in Ash1l^+/+^ and Ash1l^+/GT^ mice. As shown in Figure 4C–E, *Ash1l* haploinsufficiency significantly decreased rheobase (Ash1l^+/+^: 71 ± 4.2 pA; Ash1l^+/GT^: 42.5 ± 3.2 pA. *t*_38_ = 5.4, *p* < 0.001) and shortened first spike latency (Ash1l^+/+^: 187.4 ± 7.7 ms; Ash1l^+/GT^: 132.3 ± 9.7 ms. *t*_38_ = 4.4, *p* < 0.001). The thresholds of action potentials were significantly lower in the pyramidal neurons from Ash1l^+/GT^ mice (Ash1l^+/+^: −44.8 ± 0.6 mV; Ash1l^+/GT^: −50.3 ± 0.4 mV. *t*_38_ = 7.3, *p* < 0.001). These results demonstrated that *Ash1l* haploinsufficiency increased intrinsic excitability of pyramidal neurons in the PFC. There were no differences in resting membrane potentials, cell capacitance, input resistance, and membrane time constant between Ash1l^+/+^ mice and Ash1l^+/GT^ mice (Figure 4F–I), suggesting that *Ash1l* haploinsufficiency had no effects on the passive membrane properties of pyramidal neurons, which are highly related to the ability of neurons to generate action potentials [49].

Further, we examined the frequency of spontaneous action potentials (sAPs) of pyramidal neurons to determine the impact of *Ash1l* haploinsufficiency on neuronal excitability driven by synaptic inputs. As shown in Figure 5A,B, the frequency of sAPs was significantly increased in Ash1l^+/GT^ mice, compared with Ash1l^+/+^ mice (Ash1l^+/+^: 1.1 ± 0.06 Hz; Ash1l^+/GT^: 1.7 ± 0.07 Hz. n = 20 neurons/2 male and 3 female mice/group. *t*_38_ = 6.1, *p* < 0.001, unpaired two-tailed *t*-test). To find out the physiological basis of the hyperactivity of pyramidal neurons in Ash1l^+/GT^ mice, we further examined excitatory and inhibitory synaptic transmission by measuring AMPA (α-amino-3-hydroxy-5-methyl-4-isoxazolepropionic acid) receptor-mediated sEPSC and GABA_A_ (g-aminobutyric acid) receptor-mediated sIPSC. As shown in Figure 5C–H, *Ash1l* haploinsufficiency significantly increased the frequency but not amplitude of sEPSC in male and female mice (Amplitude: Ash1l^+/+^: 16.2 ± 0.8 pA; Ash1l^+/GT^: 16.2 ± 0.8 pA. *t*_38_ = 0.01, *p* = 0.99; Frequency: Ash1l^+/+^: 1.2 ± 0.08 Hz; Ash1l^+/GT^: 1.9 ± 0.18 Hz. n = 20 neurons/2 male and 3 female mice/group. *t*_38_ = 6.1, *p* < 0.001, unpaired two-tailed *t*-test). *Ash1l* haploinsufficiency significantly decreased the amplitude and frequency of sIPSC in male and female Ash1l^+/GT^ mice (Amplitude: Ash1l^+/+^: 33.8 ± 1.2 pA; Ash1l^+/GT^: 26.4 ± 1.3 pA. *t*_38_ = 4.3, *p* < 0.001; Frequency: Ash1l^+/+^: 3.8 ± 0.13 Hz; Ash1l^+/GT^: 2.5 ± 0.14 Hz. n = 20 neurons/2 male and 3 female mice/group. *t*_38_ = 6.6, *p* < 0.001, unpaired two-tailed *t*-test). These results demonstrated that *Ash1l* haploinsufficiency dampened pre- and post-synaptic GABAergic inhibition, and enhanced pre-synaptic glutamatergic excitation, which increased the excitability of pyramidal neurons in the PFC.

### 3.5. Ash1l Haploinsufficiency Alters the Transcriptional Levels of the Key Excitatory and Inhibitory Synaptic Genes in the PFC

To determine the molecular mechanisms underlying the synaptic dysfunction caused by *Ash1l* haploinsufficiency, we examined the transcriptional levels of the key excitatory and inhibitory synaptic genes by qPCR. As shown in Figure 6, the mRNA level of excitatory synaptic genes *Grin2a/b* (encoding NR2A/B) and *Grm2/3* (encoding mGluR2/3) were significantly decreased in Ash1l^+/GT^ mice (*Grin2a*: Ash1l^+/+^: 1.0 ± 0.04; Ash1l^+/GT^: 0.75 ± 0.03 pA. *t*_22_ = 4.7, *p* < 0.001; *Grin2b*: Ash1l^+/+^: 1.0 ± 0.03; Ash1l^+/GT^: 0.76 ± 0.03. *t_22_* = 6.0, *p* < 0.001; *Grm2*: Ash1l^+/+^: 1.0 ± 0.03; Ash1l^+/GT^: 0.73 ± 0.02. *t*_22_ = 8.0, *p* < 0.001; *Grm3*: Ash1l^+/+^: 1.0 ± 0.02; Ash1l^+/GT^: 0.76 ± 0.02. *t*_22_ = 8.4, *p* < 0.001. n = 12 mice (6 males and 6 females)/group, unpaired two-tailed *t*-test), while *Grin1* (encoding NMDA receptor NR1 subunit) and *Gria1/2* (encoding AMPA receptor GluR1/2 subunits) were largely unchanged. The inhibitory synaptic genes *Gabra1/b1/g2* (encoding GABA_A_ receptor α1, β1 and γ2 subunits), *Pvalb* (encoding Parvalbumin) and *Sst* (encoding Somatostatin) were also significantly decreased in Ash1l^+/GT^ mice (*Gabra1*: Ash1l^+/+^: 1.0 ± 0.05; Ash1l^+/GT^: 0.74 ± 0.04. *t*_22_ = 4.1, *p* < 0.001; *Gabrb1*: Ash1l^+/+^: 1.0 ± 0.05; Ash1l^+/GT^: 0.65 ± 0.03. *t_22_* = 6.1, *p* < 0.001; *Gabrg2*: Ash1l^+/+^: 1.0 ± 0.04; Ash1l^+/GT^: 0.67 ± 0.05. *t_22_* = 4.9, *p* < 0.001; *Pvalb*: Ash1l^+/+^: 1.0 ± 0.09; Ash1l^+/GT^: 0.68 ± 0.04. *t*_22_ = 3.4, *p* < 0.01; *Sst*: Ash1l^+/+^: 1.0 ± 0.07; Ash1l^+/GT^: 0.67 ± 0.05. *t*_22_ = 3.9, *p* < 0.001; n = 12 mice (6 males and 6 females)/group, unpaired two-tailed *t*-test), while *Gabrb2* (encoding GABA_A_ receptor β2) was not changed.

### 3.6. Chemogenetic Inhibition of Pyramidal Neurons in the PFC Ameliorates Autism-Like Social Deficits and Abolishes Absence-Like Seizures in Ash1l Haploinsufficiency Mice

Activating hM4Di DREADDs with CNO decreases neuronal excitability by inducing membrane hyperpolarization [50]. To determine the rescuing effect of Gi-DREADD-induced inhibition of PFC pyramidal neurons in *Ash1l* haploinsufficiency mice, we injected mCherry-tagged CaMKII-driven Gi-DREADD adeno-associated virus (AAV) bilaterally into the medial PFC, which facilitates Gi-DREADD expression in the pyramidal neurons (Figure 7A). As shown in Figure 7B–D, CNO (3 mg/kg, i.p.)-treated Ash1l^+/GT^ mice (hM4Di-injected) spent significantly more time exploring the social stimulus and increased social preference index (Ash1l^+/GT^ + Gi + saline: social: 95.6 ± 5.5 s, nonsocial: 83.1 ± 5.3 s; Ash1l^+/GT^ + Gi + CNO: social: 145.7 ± 6.9 s, nonsocial: 56.8 ± 4.0 s, n = 12 mice (6 males and 6 females)/group; *F*
_Treatment (1, 44)_ = 4.7, *p* = 0.036; *F*
_Soc vs. NS (1, 44)_ = 85, *p* < 0.001, *F*
_Interaction (1, 44)_ = 48.1, *p* < 0.001; two-way ANOVA; Social preference index: Ash1l^+/GT^ + Gi + saline: 7.2% ± 4.4%; Ash1l^+/GT^ + Gi + CNO: 44.1% ± 2.3%. n = 12 mice (6 males and 6 females)/group. *p* < 0.001, unpaired two-tailed *t*-test), indicating the dramatic amelioration of autism-like social deficits. CNO treatment terminated epileptiform discharges on EEG in Ash1l^+/GT^ mice (hM4Di-injected), which indicated that the absence-like seizures were abolished. The increased delta frequency and decreased gamma frequency were restored in Ash1l^+/GT^ mice (hM4Di-injected) after CNO treatment (Figure 7E,F). These results demonstrated that *Ash1l* haploinsufficiency-induced autism-like social deficits and absence-like seizures could be mitigated by inhibition of pyramidal neuronal activity in the PFC.

## 4. Discussion

Valid animal models are essential for understanding the pathophysiology of ASD and epilepsy [51]. *ASH1L*, located at chromosomal band 1q22, has been identified as a high-risk gene for ASD, which is mainly involved in the regulation of gene expression and neuronal communication [8,13]. In this study, we used a germline *Ash1l* knockout mouse model and found that *Ash1l* haploinsufficiency causes autism-like behavioral deficits, absence-like seizures, and increases the susceptibility for PTZ-induced convulsive seizures, confirming the causal role of *ASH1L* in ASD and epilepsy.

The clinical manifestations caused by *ASH1L* mutations are extensive phenotypic heterogeneity, such as intellectual disability, autosomal dominant 52 (MRD) [13,14] and Tourette syndrome [15], suggesting the common genetic risk of *ASH1L* for clinically distinct syndromes. Social deficits and repetitive and restrictive interests or activities are two core symptoms of ASD. Male and female Ash1l^+/GT^ mice displayed social deficits and increased self-grooming time, which recapitulates the phenotype of some autistic children carrying *ASH1L* variants and is consistent with the previous report that mice with conditional knockout *Ash1l* in the forebrain and global *Ash1l* haploinsufficiency displayed autistic-like behaviors [16,17]. Seizures have been observed in children carrying *ASH1L* mutations [52,53,54]. One case report showed a novel heterozygous mutation (c.2678dup/p.Lys894*) in *ASH1L* identified in twin sisters who initially displayed spike-wave discharges observed on EEG, absence seizures, and further developed generalized tonic–clonic seizures [54]. Here, we found *Ash1l* haploinsufficiency caused absence-like seizures, and repeated injections of low-dose PTZ significantly increased the risk for convulsive seizures in Ash1l^+/GT^ mice.

H3K4me3 is an epigenetic modification to the DNA packaging protein histone H3, which plays a significant role during early embryonic development and ASD [55,56]. Neuronal-specific H3K4me3 peaks are enriched in synaptic transmission components in human PFC during early postnatal development [57]. ASH1L plays a fundamental role in chromatin architecture and gene expression regulation via catalyzing H3K4 and H3K36 methylation [58,59]. Our previous studies showed that *Ash1l* severe deficiency in the PFC significantly decreased H3K4me3, which indicates that the *ASH1L*-related syndrome is likely linked to synaptic dysfunction in the PFC [18]. The significantly elevated eAP, sAP, increased frequency of sEPSC, and reduced amplitude and frequency of sIPSC in pyramidal neurons of Ash1l^+/GT^ mice indicated *Ash1l* haploinsufficiency altered intrinsic neuronal properties and disrupted synaptic excitation and inhibition, leading to a subsequent neural hyperexcitability in the PFC. Our previous transcriptomic analyses identified that downregulated genes by *Ash1l* severe deficiency are enriched in synaptic homeostasis and ion channels. Real-time PCR analyses are consistent with altered expressions of excitatory and inhibitory synaptic genes in idiopathic human ASD patients [18,60], which provides a mechanism driving the phenotypes in humans carrying *ASH1L* variants.

The increased excitability of pyramidal neurons in the PFC by *Ash1l* haploinsufficiency prompted us to target pyramidal neurons for therapeutic intervention. Using the DREADD-based strategy, we found that chemogenetic inhibition of pyramidal neurons in the PFC ameliorated autism-like social deficits and abolished absence-like seizures in Ash1l^+/GT^ mice. This suggests that manipulating neuronal excitability could be an effective way to treat ASD-related social deficits and epilepsy.

In summary, *Ash1l* haploinsufficiency mice generated by a gene-trap strategy exhibit a range of behavioral phenotypes associated with *ASH1L* mutations. Elevated excitability of pyramidal neurons in the PFC is strongly implicated in autism-like behavioral deficits and absence-like seizures in Ash1l^+/GT^ mice. Chemogenetic inhibition of PFC is sufficient to ameliorate autism-like social deficits and abolish absence-like seizures in Ash1l^+/GT^ mice, which reveal potential therapeutic strategies for the treatment of *ASH1L*-associated ASD and epilepsy.

## Figures and Tables

**Figure 1 genes-15-01619-f001:**
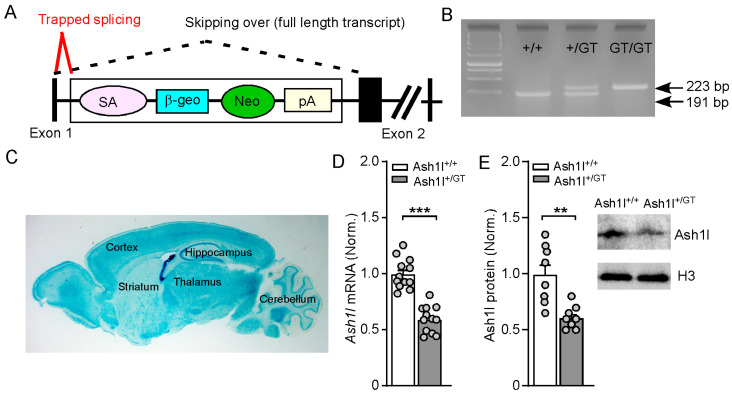
Validation of gene-trap knockout of *Ash1l* in the brain. (**A**) A schematic diagram showing gene-trap cassette insertion into the intron 1 of the *Ash1l* gene. SA: splicing acceptor; β-geo: β-galactosidase; Neo, neomycin; pA: polyadenylation sequence. (**B**) A representative PCR analysis showing the genotyping of mice with indicated genotypes. (**C**) A representative image showing LacZ signals in the whole brain from a Ash1l^+/GT^ mouse. (**D**) Quantitative PCR showing *Ash1l* mRNA levels in PFC of Ash1l^+/+^ and Ash1l^+/GT^ mice. *** *p* < 0.001, unpaired two-tailed *t*-test. n = 12 mice (6 males and 6 females)/group. (**E**) Western blot showing Ash1l protein levels in PFC of Ash1l^+/+^ and Ash1l^+/GT^ mice. ** *p* < 0.01, unpaired two-tailed *t*-test. n = 8 mice (4 males and 4 females)/group.

**Figure 2 genes-15-01619-f002:**
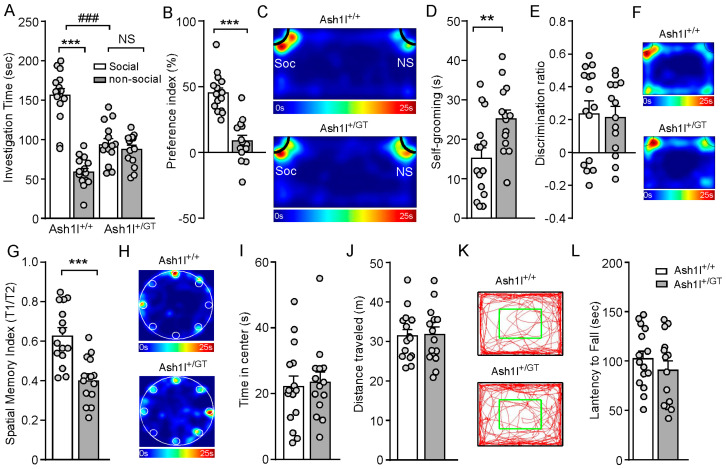
*Ash1l* haploinsufficiency causes autism-like behavioral deficits in male and female mice. Bar graphs showing the investigation time (**A**) and social preference index (**B**) in the three-chamber sociability test. A: *** *p* < 0.001, Soc versus NS; ### *p* < 0.001, Ash1l^+/GT^ versus Ash1l^+/+^ mice. B: *** *p* < 0.001, Ash1l^+/GT^ versus Ash1l^+/+^ mice. (**C**) Representative heatmaps illustrating the time spent in different locations during social preference test. (**D**) Bar graphs showing the self-grooming time in Ash1l^+/+^ and Ash1l^+/GT^ mice. ** *p* < 0.01, Ash1l^+/GT^ versus Ash1l^+/+^ mice. (**E**) Bar graphs showing the discrimination index and representative heatmaps (**F**) showing the time spent exploring the familiar and novel object during the NOR test. (**G**) Bar graphs showing the spatial memory index (T1/T2) and representative heatmaps (**H**) illustrating the time spent in different locations of the arena in Barnes maze test. *** *p* < 0.001, Ash1l^+/GT^ versus Ash1l^+/+^ mice. (**I**,**J**) Bar graphs and representative trajectory diagrams (**K**) showing time spent in center and total distance traveled during open field test. (**L**) Bar graphs showing the latency to fall in the rotarod test. Ash1l^+/+^ mice: n = 15 mice (7 males and 8 females); Ash1l^+/GT^ mice: n = 14 mice (7 males and 7 females).

**Figure 3 genes-15-01619-f003:**
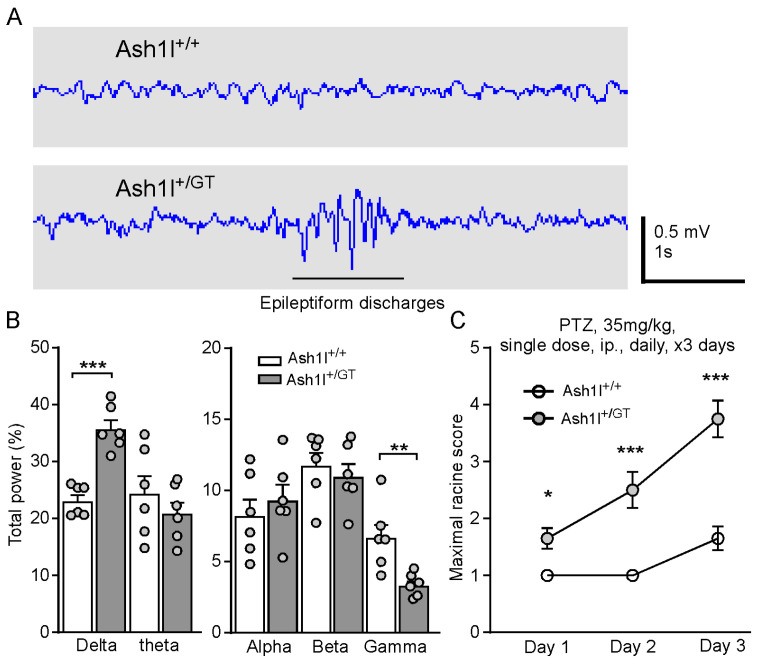
*Ash1l* haploinsufficiency causes absence-like seizures and increases the susceptibility for PTZ-induced convulsive seizures. (**A**) Representative EEG recordings showing the neural activity in the PFC of freely moving Ash1l^+/+^ and Ash1l^+/GT^ mice. (**B**) Comparison of percentage of total power in each EEG frequency band between Ash1l^+/+^ and Ash1l^+/GT^ mice. EEG band: Delta (0.1–4 Hz), Theta (4–8 Hz), Alpha (8–13 Hz), Beta (13–30 Hz), and Gamma (30–60 Hz). ** *p* < 0.01, *** *p* < 0.001, unpaired two-tailed *t* test, Ash1l^+/GT^ versus Ash1l^+/+^ mice. n = 6 mice (3 males and 3 females)/group. (**C**) Plots showing the Racine score of seizure activity in Ash1l^+/+^ and Ash1l^+/GT^ mice induced by PTZ. * *p* < 0.05, *** *p* < 0.001, Ash1l^+/GT^ versus Ash1l^+/+^ mice. n = 20 mice (10 males and 10 females)/group, two-way ANOVA.

**Figure 4 genes-15-01619-f004:**
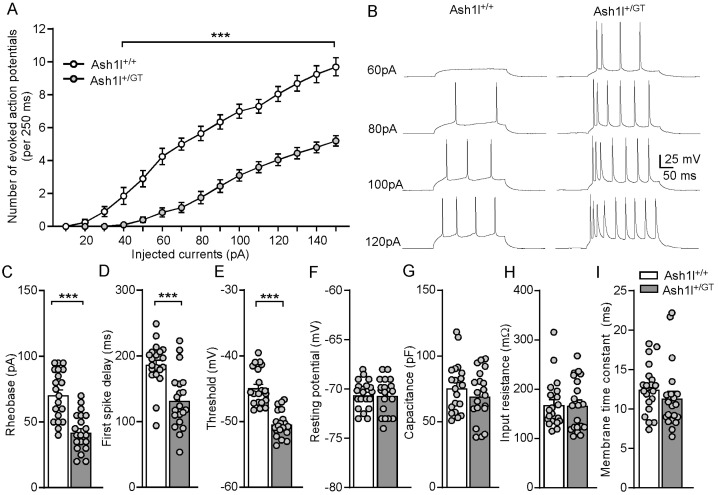
*Ash1l* haploinsufficiency significantly increases the intrinsic excitability of pyramidal neurons in the PFC of male and female mice. (**A**) Quantification of the number of evoked action potentials in the pyramidal neurons from Ash1l^+/+^ and Ash1l^+/GT^ mice. *** *p* < 0.001, two-way ANOVA. n = 20 neurons/2 male and 3 female mice/group. (**B**) Representative action potential traces. Bar graphs showing rheobase (**C**), first spike delay (**D**), threshold (**E**), resting membrane potential (**F**), capacitance (**G**), input resistance (**H**), and membrane time constant (**I**) in the pyramidal neurons from Ash1l^+/+^ and Ash1l^+/GT^ mice. *** *p* < 0.001, unpaired two-tailed *t*-test. n = 20 neurons/2 male and 3 female mice/group.

**Figure 5 genes-15-01619-f005:**
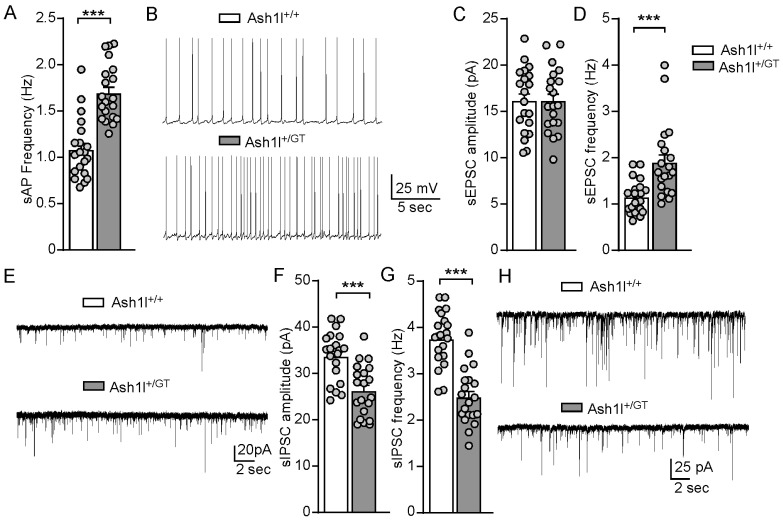
*Ash1l* haploinsufficiency elevates the balance of excitatory and inhibitory synaptic transmission. (**A**) Bar graphs showing the frequency of synaptic-driven sAP in pyramidal neurons of PFC from Ash1l^+/+^ and Ash1l^+/GT^ mice. *** *p* < 0.001, unpaired two-tailed *t*-test. n = 20 neurons/2 male and 3 female mice/group. (**B**) Representative sAP traces. Bar graphs of spontaneous EPSC amplitude (**C**) and frequency (**D**) in pyramidal neurons of PFC from Ash1l^+/+^ and Ash1l^+/GT^ mice. *** *p* < 0.001, unpaired two-tailed *t*-test. n = 20 cells/2 male and 3 female mice/group. (**E**) Representative sEPSC traces. Bar graphs of spontaneous IPSC amplitude (**F**) and frequency (**G**) in pyramidal neurons of PFC from Ash1l^+/+^ and Ash1l^+/GT^ mice. *** *p* < 0.001, unpaired two-tailed *t*-test. n = 20 neurons/2 male and 3 female mice/group. (**H**) Representative sIPSC traces.

**Figure 6 genes-15-01619-f006:**
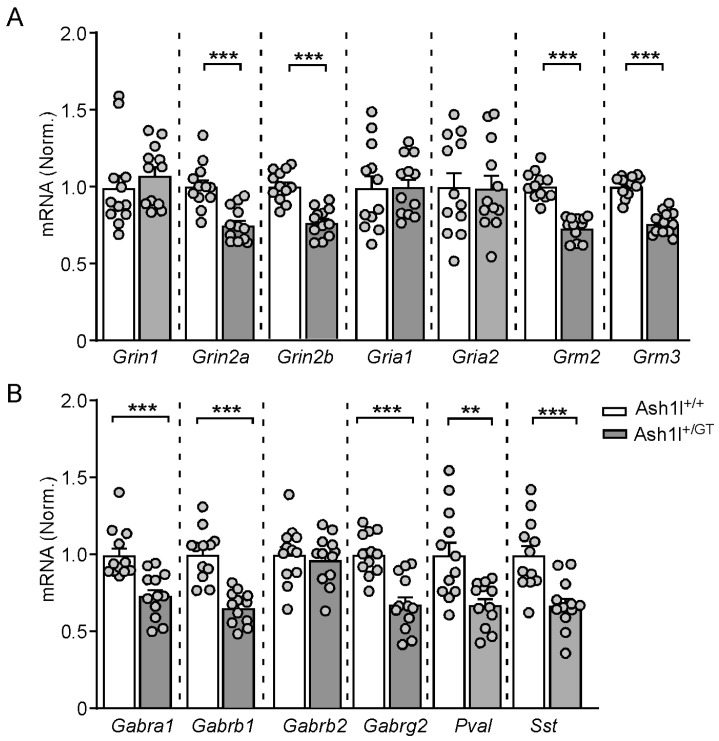
*Ash1l* haploinsufficiency alters the transcriptional levels of the key synaptic genes in the PFC of Ash1l^+/+^ and Ash1l^+/GT^ mice. Quantitative real-time PCR showing the transcriptional level of the key excitatory (**A**) and inhibitory (**B**) synaptic genes in the PFC of Ash1l^+/+^ and Ash1l^+/GT^ mice. ** *p* < 0.01, *** *p* < 0.001, unpaired two-tailed *t*-test. n = 12 mice (6 males and 6 females)/group.

**Figure 7 genes-15-01619-f007:**
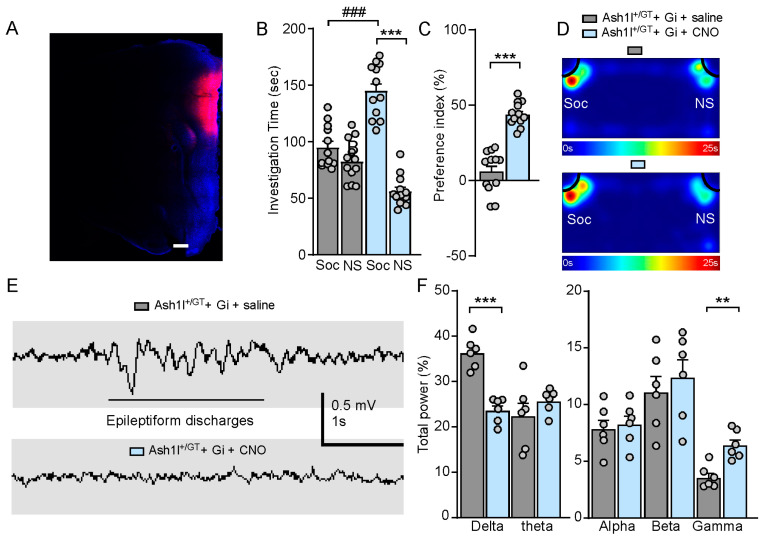
Chemogenetic inhibition of PFC ameliorates autism-like social deficits and abolishes absence-like seizures in Ash1l^+/GT^ mice. (**A**) A confocal image showing the expression of Gi-DREADD in the medial PFC (stained with DAPI, blue) from a Ash1l^+/GT^ mouse. Scale bar: 300 µm. Bar graphs showing the time spent investigating social (Soc) and non-social (NS) stimulus (**B**) and social preference index (**C**) in the three-chamber sociability test of Ash1l^+/GT^ mice (Gi-DREADD) treated with saline or CNO. A: *** *p* < 0.001, Soc versus NS; ### *p* < 0.001, CNO versus saline. B: *** *p* < 0.001, CNO versus saline. n = 12 mice (6 males and 6 females)/group. (**D**) Representative heatmaps illustrating the time spent in different locations from the social preference test. (**E**) Representative EEG recordings showing chemogenetic inhibition of PFC abolishes epileptiform discharges in freely moving Ash1l^+/GT^ mice infected with Gi-DREADD. (**F**) Comparison of percentage of total power in each EEG frequency band between Ash1l^+/GT^ mice (Gi-DREADD) treated with saline or CNO. EEG band: Delta (0.1–4 Hz), Theta (4–8 Hz), Alpha (8–13 Hz), Beta (13–30 Hz), and Gamma (30–60 Hz). ** *p* < 0.01, *** *p* < 0.001, CNO versus saline, unpaired two-tailed *t*-test. n = 6 mice (3 males and 3 females)/group.

**Table 1 genes-15-01619-t001:** List of primers used in qPCR experiments.

Target Gene	Forward	Reverse	Gene Reference	Length (bp)
*Gapdh*	gacaactcactcaagattgtcag	atggcatggactgtggtcatgag	NM_001289726.1	122
*Ash1l*	tgggaagatgacagatgaga	aagatggatgctttcttcgg	NM_138679.5	121
*Grin1*	catcggacttcagctaatca	gtccccatcctcattgaatt	NM_008169.3	238
*Grin2a*	ggctacagagacttcatcag	atccagaagaaatcgtagcc	NM_008170.4	233
*Grin2b*	ttaacaactccgtacctgtg	tggaacttcttgtcactcag	NM_008171.4	175
*Gria1*	gccttaatcgagttctgcta	gaatggattgcatggacttg	NM_008165.4	205
*Gria2*	agcctatgagatctggatgt	gagagagatcttggcgaaat	NM_001083806.3	228
*Grm2*	gcttaggttcctggcact	ttaacaggtccacactcctc	NM_001160353	150
*Grm3*	caattacttgcttccaggag	tagtcaacgatgctctgaca	NM_181850	110
*Gabra1*	caccatgaggttgaccgtga	ctacaaccactgaacgggct	NM_010250.5	158
*Gabrb1*	catagacatggtctcggaag	gtcagctactctgttgtcaa	NM_008069	130
*Gabrb2*	atttggtggctcaaacggtc	gagatttcctcaccagcagga	NM_008070.4	168
*Gabrg2*	ggagccggcatcaaatcatc	cttttggcttgtgaagcctgg	NM_008073.4	214
*Pvalb*	ggtgaagaaggtgttccata	cagacaagtctctggcatct	NM_001330686	110
*Sst*	cagactccgtcagtttctgc	atcattctctgtctggttgg	NM_009215.1	112

## Data Availability

The original contributions presented in this study are included in this article, further inquiries can be directed to the corresponding author.

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
