# Peer review of "Chemogenetic Inhibition of Prefrontal Cortex Ameliorates Autism-Like Social Deficits and Absence-Like Seizures in a Gene-Trap Ash1l Haploinsufficiency Mouse Model"

_genes, 2024, doi:10.3390/genes15121619_

Round 1

Reviewer 1 Report

Comments and Suggestions for Authors

This is a very nice piece of work. The experiments are presented in a straightforward manner and clearly described throughout the manuscript.

I only have some minor comments I would the authors to address:

- The authors should include a picture of a mouse brain showing the expression of Ashl1 (mRNA or protein) in WT animals. This could be placed in Figure 1 together with Figure 1C.

- Please provide information about the amplicon length of the identified primers (Table 1). If primers were chosen from the literature, please add the reference. If primers were designed by the authors, were they exon-exon spanning or not?

- Provide a picture of the mouse PFC to visualise the expression of the virus injected instead of a schematic representation in Figure 7A.

Author Response

We appreciate the reviewer's constructive comments and invaluable suggestions on this paper. A point-by-point response to the comments is attached below. All changes in the manuscript text file are highlighted in blue. We hope that the revised paper is suitable to be published in Genes. Thanks for your kind consideration!

Q: The authors should include a picture of a mouse brain showing the expression of Ashl1 (mRNA or protein) in WT animals. This could be placed in Figure 1 together with Figure 1C.

A: Since there was no gene-trap cassette in the Ash1l+/+ mice, ß-gal staining was negative in Ash1l+/+ mice. Given that the brain slice of Ash1l+/+ mice was transparent, we did not add it in Figure 1. Please see the image in PDF file.

Q: Please provide information about the amplicon length of the identified primers (Table 1). If primers were chosen from the literature, please add the reference. If primers were designed by the authors, were they exon-exon spanning or not?

A: We added the amplicon length in the table. The primers were designed by us, which were used in our previous publications. Since the primers for real-time PCR were relatively short, it is difficult to find the primers spanning exon-exon. These primers did not span exon-exon.

Q: Provide a picture of the mouse PFC to visualize the expression of the virus injected instead of a schematic representation in Figure 7A.

A: We updated Figure 7A as the reviewer suggested.

Reviewer 2 Report

Comments and Suggestions for Authors

The authors investigated the consequences of the Ash1l deficiency in mice in showing autism-like behaviors tested with standard methods. Previosuly same authors showed an association with epileptic seizures in mice. Issues related to authistic behavious are linked to overactive pyramidal neurons in the prefrontal cortex (PFC). Turning down the activity in this brain region with chemogenetics helps fix social behavior problems and stops the seizures. This points to possible ways to treat autism and epilepsy linked to ASH1L in humans.

The manuscript is interesting and well-written and the authors have extensive knowledge of the topic backed up with previous studies. Figures are illustrative, the methodology is detailed and sophisticated (e.g. patch clamp).

Major comment:

iThanticare is 51%, and some parts are very similar to another manuscript by the same authors. Please rephrase

Minor comments:

A table with primers could be in the supplement

Are there any other, apart from seizures, typical phenotypic features in humans with likely pathogenic variants in ASH1L?

Author Response

We appreciate the reviewer's constructive comments and invaluable suggestions on this paper. A point-by-point response to the comments is attached below. All changes in the manuscript text file are highlighted in blue. We hope that the revised paper is suitable to be published in Genes. Thanks for your kind consideration!

Q: iThanticare is 51%, and some parts are very similar to another manuscript by the same authors. Please rephrase.

A: We use standard methods in our lab, which may cause iThanticare is 51% in paper.

Q: A table with primers could be in the supplement

A: To facilitate the readers to check or use the primers, we decide to keep the primers in the text.

Q: Are there any other, apart from seizures, typical phenotypic features in humans with likely pathogenic variants in ASH1L?

A: The clinical manifestations caused by ASH1L mutations are extensive phenotypic heterogeneity. Please see the information in discussion (lines 644-647).

Reviewer 3 Report

Comments and Suggestions for Authors

Summary
The authors conducted a comprehensive study investigating ASH1L haploinsufficiency in mouse models to elucidate its strong correlation with ASD and epileptic seizures. The research is well-executed, and the results are clearly presented. This research will pave the way for the development of a specific therapy.

Title
Ensure that the title adheres to MDPI’s Instructions for Authors, using capital letters for each significant word.

Abstract
The abstract is well-written and effectively communicates the aim and significance of this valuable work.

Introduction
The OMIM annotation for “Intellectual Developmental Disorder, Autosomal Dominant 52” (MIM phenotype number #617796) should be referenced in the introduction. Additionally, this aspect should be revisited in the Discussion section to identify potential discrepancies or similarities with the clinical synopsis detailed in the OMIM database. https://www.omim.org/entry/607999?search=ASH1L&highlight=ash1l

Materials and Methods
This section is thoroughly detailed. However, I recommend moving Table 1 to this section for better contextual alignment.

Results
No major concerns.

Discussion

  • Line 643: Verify the usage of the symbol “&” for correctness and consistency.
  • Include a comparative analysis of the phenotypic findings with the clinical synopsis mentioned earlier, as this could enhance the discussion of your results.

Author Response

We appreciate the reviewer's constructive comments and invaluable suggestions on this paper. A point-by-point response to the comments is attached below. All changes in the manuscript text file are highlighted in blue. We hope that the revised paper is suitable to be published in Genes. Thanks for your kind consideration!

Q: Title: Ensure that the title adheres to MDPI’s Instructions for Authors, using capital letters for each significant word.

A: We updated the titles.

Q: Introduction: The OMIM annotation for “Intellectual Developmental Disorder, Autosomal Dominant 52” (MIM phenotype number #617796) should be referenced in the introduction. Additionally, this aspect should be revisited in the Discussion section to identify potential discrepancies or similarities with the clinical synopsis detailed in the OMIM database. https://www.omim.org/entry/607999?search=ASH1L&highlight=ash1l

A: We included “Intellectual Developmental Disorder, Autosomal Dominant 52” into other neurodevelopmental deficits (lines 46-47). We added additional references #14 and #15.

Q: I recommend moving Table 1 to this section for better contextual alignment.

A: We did.

Q: Discussion Line 643: Verify the usage of the symbol “&” for correctness and consistency.

A: We did.

Q: Include a comparative analysis of the phenotypic findings with the clinical synopsis mentioned earlier, as this could enhance the discussion of your results.

A: We added in the discussion (lines 644-647).

Round 2

Reviewer 2 Report

Comments and Suggestions for Authors

Dear authors,

Thanks for your answer. 

I would accept the manuscript.